# A Virtual Exercise Program throughout Pregnancy during the COVID-19 Pandemic Modifies Maternal Weight Gain, Smoking Habits and Birth Weight—Randomized Clinical Trial

**DOI:** 10.3390/jcm11144045

**Published:** 2022-07-13

**Authors:** Cristina Silva-Jose, Miguel Sánchez-Polán, Ruben Barakat, Ángeles Díaz-Blanco, Michelle F. Mottola, Ignacio Refoyo

**Affiliations:** 1AFIPE Research Group, Faculty of Physical Activity and Sport Sciences-INEF, Universidad Politécnica de Madrid, 28040 Madrid, Spain; cristina.silva.jose@upm.es (C.S.-J.); rubenomar.barakat@upm.es (R.B.); 2Gynecology and Obstetrics Department, Hospital Universitario Severo Ochoa de Leganés, 28911 Leganés, Spain; mariycuco@yahoo.es; 3R. Samuel McLaughlin Foundation-Exercise and Pregnancy Lab, School of Kinesiology, Faculty of Health Sciences, Department of Anatomy & Cell Biology, Schulich School of Medicine & Dentistry, Children’s Health Research Institute, The University of Western Ontario London, London, ON N6A 3K7, Canada; mmottola@uwo.ca; 4Sports Department, Faculty of Physical Activity and Sports Sciences-INEF, Universidad Politécnica de Madrid, 28040 Madrid, Spain; ignacio.refoyo@upm.es

**Keywords:** exercise, pregnant, epigenetics, physical activity, integral wellbeing, smoking

## Abstract

The intrauterine environment is key to health from a short- and long-term perspective. Birth weight is an important indicator that may influence the fetal environment due to epigenetics. Considering physical inactivity, in parallel with higher levels of stress, affecting smoking patterns and the physical and emotional health of the pregnant population, maintaining the health of future generations is crucial. A randomized clinical trial (NCT04563065) was conducted. One-hundred and ninety-two healthy pregnant individuals were assigned to the intervention (IG) or control (CG) group. Overall, significant differences were found between groups when stratified by birth weight (χ^2^ (1) = 6.610; *p* = 0.037) with low birth weight and macrosomia found more often in the CG (4% vs. 14% and 3% vs. 9%, respectively) and higher admissions to the neonatal intensive care unit (χ^2^ (1) = 5.075; *p* = 0.024) in the CG (20/28.6%) compared to the IG (9/13.0). Smoking during pregnancy was also found more often in the CG (12/17.1%) compared to the IG (3/4.4%) (*p* = 0.016). A virtual program of supervised exercise throughout pregnancy during the ongoing pandemic could help to maintain adequate birth weights, modify maternal smoking habits, and lower admissions to the neonatal intensive care unit.

## 1. Introduction

Birth weight (BW) is an important indicator of the intrauterine environment and maternal and newborn health [1]. Maternal body mass index (BMI), pattern of gestational weight gain and total gestational weight gain are factors that determine BW [2]. This linkage is vital in modifiable pregnancy epigenetic outcomes in relation to the future health of the newborn [3], since any alteration in the fetal environment is associated with early, and long-term implications [4,5,6]. The prevalence of childhood obesity has increased worldwide over the last three decades [7]. However, it is known that babies with appropriate BW have a lower risk of obesity, diabetes mellitus type 2 (DM2), metabolic syndrome, or reduced markers of metabolic disease in childhood or in adulthood compared to babies with excessive BW (>4000 g) [8,9]. Alarmingly, excessive BW is also associated with low to moderately elevated risks for disease such as: type 1 diabetes, hypertension, several psychiatric disorders or breast cancer during childhood [10]. Additionally, excessive birth weight increases the risk of hypoglycemia [11], shoulder dystocia (which increases the possibility of maternal hemorrhage and damage to the brachial plexus or clavicle fracture of the newborn [12]), injuries to the phrenic nerve, asphyxia, or meconium aspiration syndrome [13], thus increasing the risk of maternal or infant mortality. Moreover, having inadequate birth weight (<2500 g) is related to developmental complications in the uterine environment, nutritional deficiencies, and premature growth that can lead to hypoglycemia after childbirth and risk of insulin resistance from the first year of life [14]. This inadequate weight could lead to chronic diseases increasing the risk of coronary heart disease, cerebrovascular incidents, and DM2 [4,15,16]. Previous studies also link low birth weight with a decrease in brain volume that would lead to negative neurocognitive development [17,18,19,20].

In addition to maternal weight management, another interrelated and significant lifestyle factor that can influence fetal, neonatal and child well-being and growth is maternal smoking during pregnancy [21]. Smoking cessation during pregnancy improves maternal and child outcomes [22]. However, only approximately 25% of pregnant smokers stop for their pregnancy, and half to two-thirds of them return to smoking after birth [23]. Maternal smoking may result in persistent alterations in fetal metabolism [24] and it is associated with intrauterine growth restriction, miscarriage, neonatal asphyxia [25], preterm birth, low birth weight [26,27,28,29], congenital abnormalities and neonatal or sudden infant death [27,30], and conversely, an increased risk of overweight and obesity in childhood by 50% [31] and 34–41% during youth and adulthood [32,33]. According to The Developmental Origins of Health and Disease (DOHaD) theory [34], the fetal period has enormous plasticity and the ability to respond to lifestyle and the maternal environment [35], adapting to external signals such as alcohol consumption [36,37] or cannabis smoking [38] that negatively affect BW. In addition, the parental diet could influence the epigenetic role, and imprinting in the offspring [39], which may alter neonatal weight and composition, affecting future generations [40].

Maternal exercise has been shown to decrease BW [41], which helps to decrease comorbidities related to chronic disease risk [2] and reduce percentages of fat mass in both mothers and infants [17]. Recent research has also suggested a possible epigenetic impact on the offspring [40,42]. Similarly, physical activity (PA) programs might reduce the intensity of urges to smoke, which are the main cause of smoking relapse [43,44], and PA can be used as a strategy together with behavioral support [22]. Furthermore, findings suggest that the children of trained mothers have greater neurodevelopment, cognitive and general intelligence [45], especially in oral language, and a greater sense of orientation and self-control [46]. More research is necessary to confirm these findings [47]. Current research on public health complications due to the continuity of limitations and social isolation caused by the Coronavirus disease 2019 (COVID-19) pandemic showed reductions in physical activity and maintenance of smoking habits, elements that increase the risk of numerous pathologies and pregnancy disorders [48,49]. One example is that a sedentary lifestyle and no physical activity may enhance anxiety and depression in expectant mothers [50] and neonatal complications [51] potentially making these conditions worse. Added to this socially rooted pattern is the difficulty in promoting and supporting PA while limiting the risk of COVID-19; it is a challenge to maintain adequate levels of PA and reduce cigarette smoking during a pandemic [52]. Therefore, novel interventions based on technological advances are good strategies during pandemic restrictions. The relationship between exercise for mothers without contraindications during pregnancy and maternal, fetal, and newborn wellbeing has been previously illustrated in the international guidelines [53,54,55]. All recommendations base an active lifestyle during pregnancy on 30 min of moderate exercise for five days a week [53,54,55], although there is a great variety among the physical exercise programs used in previous studies [56].

Therefore, during the current COVID-19 pandemic and where social distancing is recommended, the purpose of the current study was to evaluate the influence of a supervised virtual exercise program throughout pregnancy on birth weight. A secondary objective was to examine the influence of virtual exercise and the relationship between changes in the maternal environment during pregnancy (gestational weight gain or smoking) and other neonatal outcomes (neonatal intensive care unit (NICU) or mode of delivery). It was hypothesized that babies born to mothers in the intervention group would more likely have optimal birth weight (between 2500 and 4000 g) and less birth complications (NICU admissions) compared to the control group, and that excessive pregnancy weight gain and smoking during pregnancy would be modified due to the virtual exercise program.

## 2. Materials and Methods

### 2.1. Study Design

A randomized clinical trial (NCT04563065) was conducted with the Obstetrics and Gynecology Department of the Hospital Universitario Severo Ochoa (Madrid, Spain) and the Universidad Politécnica de Madrid. The current investigation was approved by the Ethical Commission of Research of the Universidad Politécnica de Madrid and the aforementioned hospital’s Ethical Commission of Clinical Research (CEIC). Participants’ personal and medical information were collected using hospital clinical records. Pregnant individuals were randomly assigned to either an intervention or a control group.

### 2.2. Participants and Randomization

A total of 371 Spanish-speaking pregnant individuals residing in Madrid were recruited and assessed for eligibility from hospital obstetric visits (Figure 1). Obstetric health providers determined the inclusion and exclusion criteria at this initial visit. Individuals between 18 and 43 years with no history or risk of preterm delivery, without multiple and complicated pregnancies, and not enrolling in any other trial or exercise program were recruited. Those who did not plan to give birth in the same hospital and who were not under medical follow-up throughout pregnancy were not included, nor were individuals with any serious medical conditions (contraindications) that prevented them from exercising safely [53,54,55]. All participants signed an informed consent form before enrolling.

REDCap software was used for the entire randomization process. Participants were randomly assigned using a 1:1 allocation ratio to either the IG or the CG. A blinded randomization sequence was made using a computer-generated list of random numbers and was uploaded to a REDCap database. Access to the REDCap software for randomization was conducted by one health provider in each hospital. All assignments were blinded to hospital staff.

### 2.3. Control Group

Pregnant individuals assigned to the CG received their routine obstetric health care at their referral hospital centres like the exercise group. In addition, they were provided with materials including physical activity, sleep habits, and smoke-free pregnancy and nutritional guidelines in different formats throughout pregnancy. To regulate their PA level, they were asked about the amount of exercise once each trimester using a “Decision Algorithm” (by telephone) [57]. If they were found to be exercising excessively, the participants were excluded from the study.

### 2.4. Intervention Group

Women enrolled in the IG received usual care at their hospitals and they were given healthy lifestyle recommendations like the control group. Participants followed a supervised virtual exercise program throughout pregnancy starting at weeks 8–10 until 39–40. The exercise program included 3 weekly sessions of 55–60 min of complementary and interrelated activities following a methodological model divided into seven, adapted to each trimester of pregnancy, previously established by our research group [58].

The exercise program was divided into two modalities that included individual and group work. The first mode consisted of one weekly session with easily recorded workouts on a list in YouTube. The second was composed of two weekly sessions supervised online through the Video Platform (Zoom Video Communications Inc., San Jose, CA, USA). Classes were provided on separate days to harmonize the participant calendars without the possibility of carrying out two sessions on the same day.

A minimum of 80% adherence was established for the final analysis of results. Control of attendance at the sessions was registered by the computer application.

The workload intensity was controlled using the values of 55–65% of maximum maternal heart rate (MHR) calculated from the Karvonen formula [59] and using an effort perception of 12–14 (Somewhat Hard) in the Borg’s Rating of Perceived Exertion Scale. During the workouts, heart rate was recorded by a heart rate monitor or for 10 s at the carotid artery and at the end of the class during a relaxation period, and participants manifested their perceived effort. Participants were informed about not overheating to prevent maternal hyperthermia. 

### 2.5. Outcomes

All data before, during, and after pregnancy for the mother, fetus and newborn were obtained by the SELENE platform that manages all hospital obstetric records. As a primary outcome, birth weight was stratified by optimal birth weight between 2500 and 4000 g, macrosomia as >4000 g and low birth weight as <2500 g [1]. Maternal data were examined by maternal age, weight, height, pre-pregnancy body mass index (BMI), parity, occupation, previous miscarriage, and smoking before and during pregnancy. Smoking status was determined from medical records. The participants were asked if they smoked prior to pregnancy (Yes/No) and at all obstetric visits (Yes/No). Smoking status during pregnancy was represented by the yes/no response at the last visit (just before labor). 

BMI was calculated as weight (kg) divided by height (m^2^), and individuals were classified as underweight (BMI < 18.5 kg/m^2^), normal weight (BMI ≥ 18.5 to 24.9 kg/m^2^), overweight (BMI ≥ 25 to 29.9 kg/m^2^), and obese (BMI ≥ 30 kg/m^2^) [60]. Gestational weight gain was calculated based on pre-pregnancy weight and weight at the last clinic visit before delivery. Gestational weight gain was classified according to the 2009 Institute of Medicine (IOM) guidelines [61]. Excessive body weight gain was determined for pre-pregnancy BMI categories for each participant; >18 kg for underweight; >16kg for normal; >11.5 kg for overweight; and >9 kg for obese. Inadequate weight gain was determined for pre-pregnancy BMI categories for each participant; <12.5 kg for underweight; <11.5 kg for normal; <7.0 kg for overweight; and <5.0 kg for obese categories [61]. Finally, childbirth delivery data included maternal gestational age, mode of delivery (non-instrumental, instrumental or c-section), type of delivery (preterm at <37 weeks of gestation), sex, birth weight, birth length, head circumference, Apgar Scores (appraise the newborn’s heart rate, muscle tone, reflexes, color, and respiratory effort) at 1 and 5 min, umbilical cord blood pH, and admission to the NICU.

### 2.6. Power Sample Calculation

Power calculations for the primary outcome (excessive birth weight) [33] were based on previous studies and used a prevalence of 10% in the intervention group and 30% in the usual care group. Under these assumptions, a two-sample comparison (χ^2^) with a 5% level of significance and a statistical power of 0.90 gave a study population of 82 participants in each group. Assuming a maximum loss to follow-up of 15%, we decided to recruit 96 participants for each of the study groups [62].

### 2.7. Statistical Analysis

Version 25.0 of IBM SPSS for Windows (IBM Corporation, Armonk, NY, USA) was used. Screening for violations of normality was checked using the Kolmogorov–Smirnov test. Pearson’s chi-square test was used to compare the type of pregnancy weight gain (inadequate, adequate, or excessive), number of participants by BMI subcategory (pre-pregnancy and final mediations), maternal smoking before and during, previous miscarriages, parity, employment occupation, mode of delivery, type of delivery, gestational age (preterm or full-term delivery), sex, and NICU admissions between the IG and CG. 

Independent *t*-tests were used to assess the differences in maternal age, weight, height, pre-pregnancy BMI, final BMI, final maternal weight, gestational weight gain, gestational age at delivery, fetal heart rate, birth weight, birth length, head circumference, weight at hospital discharge, Apgar Scores 1 and 5 min, and umbilical cord blood pH between the intervention and control groups. 

A one-way ANOVA was performed to examine birth weight differences related to maternal stratified gestational weight gain between groups. Post hoc comparisons were done with the Scheffe test.

Pairwise correlation coefficients were calculated to examine the interrelationships between maternal outcomes (smoking during pregnancy, GWG and final BMI), childbirth outcomes (mode of delivery) and newborn outcomes (birth weight, birth length, head circumference, Apgar 1, Apgar 5, cord blood pH and NICU).

Data for continuous variables are presented as means and standard deviations, and those of the nominal variables are presented as frequencies and percentages. The level of statistical significance was set at *p* < 0.05.

## 3. Results

A total of 371 women were randomized and 179 were excluded: 104 did not meet the inclusion criteria, 27 declined to participate and 48 for other reasons. Participants were divided into IG (n = 96) and CG (n = 96). In the IG, 27 pregnant women were lost to follow-up, 11 had low adherence, 7 changed hospitals, and 9 for other reasons. In the CG, 26 women were lost to follow-up: 3 had persistent bleeding, 9 changed hospitals, and 14 for other reasons. Finally, 69 individuals in the IG and 70 in the CG were analyzed (Figure 1). 

Table 1 shows the characteristics of the pregnant participants. Although smoking before pregnancy was not different between groups, significant differences were found in smoking during pregnancy (χ^2^ (1) = 5.771; *p* = 0.016).

### 3.1. Maternal Weight Gain

Table 2 shows weight-related outcomes for the pregnant participants. There were significant differences in final maternal weight (*p* = 0.003), final BMI (*p* = 0.015) and gestational weight gain (*p* = 0.048) between the groups. There also were significant differences between the IG and CG (χ^2^ (1) = 8.771; *p* = 0.032) in the type of final BMI of pregnancy.

### 3.2. Childbirth Variables

There were significant differences between the IG and CG (χ^2^ (1) = 5.972; *p* = 0.049) in mode of delivery, with C-section (12% vs. 26%) and instrumental delivery (7 vs. 12%) more often occurring in the CG. Specifically, there were significant differences between the IG and CG (χ^2^ (1) = 6.610; *p* = 0.037) in type of birth weight, with low birth weight and macrosomia found more often in the CG (4% vs. 14% and 3 % vs. 9%, respectively) and in the admission to neonatal intensive care unit (χ^2^ (1) = 5.075; *p* = 0.024) (Table 3).

Furthermore, birth weight was different within the IG depending on the type of gestational weight gain (F_(2,57)_ = 3.307; *p* = 0.035; η^2^ = 0.104). The multiple posterior comparisons made using the Scheffe test found significant differences (*p* < 0.05) between excessive and inadequate weight gain (*p* = 0.035). However, no significant differences were found within the CG (F_(2,60)_ = 1.373; *p* = 0.261; η^2^ = 0.044) (Table 4).

### 3.3. Matrix of Correlations

In both groups, final BMI was positively associated with gestational weight gain (IG: r = 0.408; *p* = 0.001; CG: r = 0.260; *p* = 0.043), birth weight with birth length (IG: r = 0.744; *p* < 0.001; CG: r = 0.728; *p* < 0.001), head circumference with birth weight (IG: r = 0.609; *p* < 0.001; CG: r = 0.725; *p* < 0.001), and birth length (IG: r = 0.743; *p* < 0.001; CG: r = 0.664; *p* < 0.001), Apgar at 1 min with Apgar at 5 min (IG: r = 0.635; *p* < 0.001; CG: r = 0.867; *p* < 0.001). Furthermore, Apgar at 1 min was negatively associated with mode of delivery (IG: r = −0.244; *p* = 0.047; CG: r = −0.304; *p* = 0.011) and NICU (IG: r = −0.296; *p* = 0.015; CG: r = −0.451; *p* < 0.001) and Apgar at 5 min was negatively associated with mode of delivery (IG: r = −0.289; *p* = 0.018; CG: r = −0.235; *p* = 0.050) and NICU (IG: r = −0.388; *p* = 0.001; CG: r = −0.494; *p* < 0.001). 

Specifically within the IG, gestational weight gain was positively correlated with smoking during pregnancy (r = 0.380; *p* = 0.002) and birth weight (r = 0.332; *p* = 0.009). Final BMI was negatively correlated with mode of delivery (r = −0.289; *p* = 0.019). On the other hand, in the CG, the mode of delivery was positively associated with smoking during pregnancy (r = 0.327; *p* = 0.006) and NICU (r = 0.381; *p* = 0.012) and pH cord blood was negatively associated with birth weight (r= −0.396; *p* = 0.015) and birth length (r = −0.484; *p* = 0.004).

## 4. Discussion

The main objective of the present study was to examine the influence of a pandemic-driven virtual structured exercise program during pregnancy on birth weight and maternal pregnancy changes (gestational weight gain or smoking) related to other neonatal outcomes (NICU admission or mode of delivery). This innovative action plan used a combination of a variety of exercises (aerobic, strength, balance, or coordination) adopting a group session structure to the online model. These findings suggest that the pregnant population can strongly join in online group workouts and continue to be active throughout pregnancy during the ongoing global pandemic.

Overall, our results showed no differences in the mean birth weight between the IG and CG like other previous studies [64,65,66]. However, when stratified by low, adequate, and high birth weight, both the extreme weight types (low birth weight and macrosomia) were seen more often in the CG (4% vs. 14% and 3% vs. 9%, respectively). It appears that being active helps to control low birth weight [67] and reduces the risk of macrosomia by up to 30% [41]. Both birth weight extremes are negatively associated with the development of the infant in the short and long term [68,69,70,71,72]. Less than half of the NICU admissions (a proxy for newborn health) were found in the IG (n = 9) compared to the CG (n = 20), with percentages for the total group being 13% vs. 29% respectively. NICU admissions have been shown to be related to extremes in birth weight [73]. In addition, the correlations showed that the NICU is associated with lower values in Apgar scores at 1 and 5 min, and specifically in the CG, with the mode of delivery being associated with C-sections to a greater extent. Therefore, the maintenance of adequate levels of physical activity throughout pregnancy and exercising according to the recommendations can avoid these complications associated with extreme birth weights [74,75].

Final pregnancy weight and weight gain were similar in each subgroup of prepregnancy BMI index, possibly due to the limited number of women in each group. Recently, RCTs have shown more uncontrolled and excessive weight gain in the less active group [1,76]. This is unfortunate since the pandemic has forced individuals to limit activity, isolate, and restrict activities, which may favor reductions in the amount of daily PA, and reduced PA levels, longer sedentary time [77], and greater weight gain, aggravating health problems and complications [78]. Not surprisingly, mode of delivery was different between groups in the present study, with non-instrumental deliveries occurring more often in the IG (81% vs. 63%) and C-sections reported more often in the CG (12% vs. 26%). These findings are also reported in the scientific literature [79,80], which appears to be related to maternal weight gain, and type of birth weight or NICU admissions [74]. 

Differences were not found in smoking habits prior to pregnancy, but they were found during pregnancy between groups. A smoke-free pregnancy is vital to both mother and child [81]. However, it was found that smoking during pregnancy was higher in the CG, with values like those found in previous general observational studies, where individuals used smoking as an agent to relieve stress [82]. Although our results do not show direct correlations with baby weight or NICU admission, other previous studies found a higher risk of having a child with low birth weight [83,84] and other negative pregnancy outcomes like spontaneous abortion [85], preterm birth or respiratory distress [86], both associated with NICU admission. The current situation caused by COVID-19 is perceived by pregnant individuals as stressful [87,88] and even though in our study all received information on healthy lifestyle (information on smoking included), the CG maintained their smoking habits. Pregnant individuals who smoked during the pandemic [89] also had lower frequencies of physical activity [90] and increased chances of anxiety and depression, as well as showing decreased success in quitting smoking [91,92]. Therefore, healthy lifestyle habits are once again recommended to ensure physical, psychological, and social well-being in this population during the COVID-19 pandemic [93].

Combining an online supervised exercise intervention with educational advice on healthy lifestyle shows improvement in maternal, fetal, and newborn health. Our RCT confirms the beneficial effects of exercise during pregnancy and demonstrates the importance of using lifestyle-focused treatments as a necessary factor for the prevention of unhealthy habits such as smoking and inappropriate and excessive maternal weight gain that could negatively affect infant and fetal development. In summary, an integrative online supervised intervention may be a relevant element in the health of the pregnant population and descendants during a pandemic.

The strengths of the present study included the variance of the exercise program, as well as the health guidelines that include tips and guidelines to avoid tobacco, within a large RCT of an online supervised aerobic exercise intervention with high adherence. Furthermore, the physical activity of the CG was measured and controlled (excluding highly active women) and these individuals received the same healthy guidelines as the IG, which avoids possible analysis biases. Finally, providing necessary information to the CG should be an important part of standard care protocols. A possible limitation of our study was the lack of nutritional evaluation. Nevertheless, all participants received standard care and information on a healthy lifestyle in obstetric care throughout the entire pregnancy. The supervision of a virtual program is not identical to a face-to-face session; however, using online technological resources allowed us to adapt to the pandemic situation and to follow the group dynamics that characterize so much of the success of the intervention. Nonetheless, in the future, we may see more online fitness programs for pregnant individuals, and therefore future studies should compare the effect of online versus face-to-face supervised group fitness classes on comprehensive maternal and fetal health. 

## 5. Conclusions

A virtual supervised exercise program throughout pregnancy during the ongoing global COVID-19 pandemic may help to maintain adequate birth weights, reducing cases of low birth weight and macrosomia, lower admissions to the neonatal intensive care unit, and promote health in infants, which may also prevent future comorbidities in both mother and infant. In addition, this program could help to promote healthy habits (stopping smoking) during pregnancy that can be maintained throughout life.

## Figures and Tables

**Figure 1 jcm-11-04045-f001:**
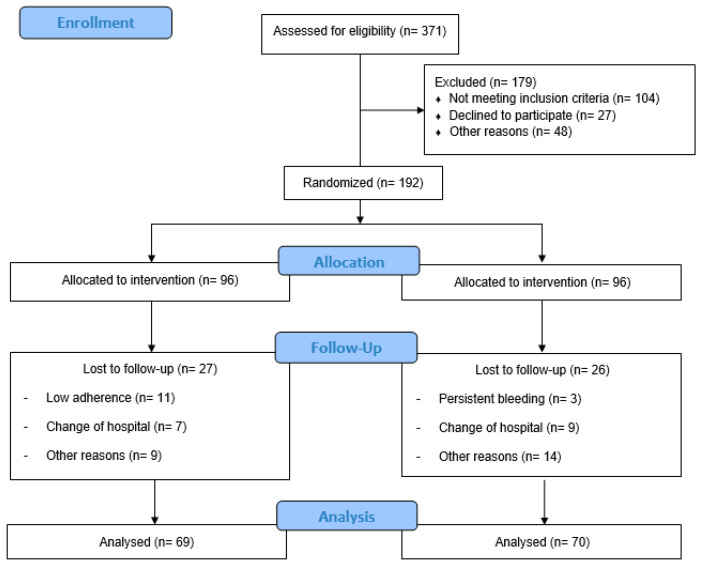
Study population flow chart.

**Table 1 jcm-11-04045-t001:** Maternal characteristics of both groups.

Maternal Characteristics			
Variable	Intervention Group (n = 69)	Control Group (n = 70)	*p*-Value
**Age (years)**	33.83 ± 3.87	33.41 ± 5.91	0.628
**Maternal height (m)**	1.64 ± 0.06	1.62 ± 0.07	0.122
**Maternal pre-pregnancy weight (kg)**	62.53 ± 9.29	66.07 ± 12.33	0.068
**Pre-pregnancy BMI (n/%)**	23.25 ± 3.26	24.75 ± 5.47	0.063
<18.5	5/7.5	3/4.8	0.081
18.5 to 24.9	46/68.7	31/50.0	
25 to 29.9	12/17.9	19/30.6	
>30	4/6.0	9/14.5	
**Parity (n/%)**			
None	54/78.3	42/60.9	0.058
One	11/15.9	23/33.3	
Two or more	4/5.8	4/5.8	
**Smoking before pregnancy**			0.284
No	42/62.7	37/53.6	
Yes	25/37.3	32/46.4	
**Smoking during pregnancy**			
No	65/95.6	58/82.9	0.016
Yes	3/4.4	12/17.1	
**Occupation (n/%)**			
Active job	25/37.9	20/29.0	0.480
Sedentary job	25/37.9	32/46.4	
Homemaker	15/22.7	17/24.6	
**Previous miscarriage (n/%)**			
None	47/68.1	43/62.3	0.156
One	18/26.1	15/21.7	
Two or more	4/5.8	11/15.9	

Data are expressed as mean ± SD, unless otherwise indicated.

**Table 2 jcm-11-04045-t002:** Maternal demographics for both groups.

	Intervention Group (n = 69)	Control Group (n = 70)	*p*-Value
**Gestational age (weeks)**	39.07 ± 1.37	38.85 ±1.45	0.771
**Final maternal weight (kg)**	73.25 ± 9.84	80.45 ± 16.29	0.003
**Final BMI index**	26.02 ± 6.56	29.23 ± 8.40	0.015
**Gestational weight gain (kg)**	9.92 ± 3.93	11.51 ± 4.96	0.048
**GWG based on pre-pregnancy BMI (kg)**			
<18.5	13.10 ± 7.53	10.50 ± 4.95	0.703
18.5 to 24.9	10.25 ± 3.64	11.70 ± 5.15	0.163
25 to 29.9	9.59 ± 2.92	11.91 ± 5.23	0.177
>30	7.40 ± 2.55	9.96 ± 3.14	0.333
**Final pregnancy BMI (n/%)**			
<18.5	3/4.5	3/4.3	0.032
18.5 to 24.9	14/21.2	11/15.9	
25 to 29.9	35/53.0	24/34.8	
>30	14/21.2	31/ 44.9	

**Table 3 jcm-11-04045-t003:** Childbirth and neonatal outcomes between IG and CG.

	Intervention Group (n = 69)	Control Group (n = 70)	*p*-Value
**Fetal heart rate (bpm)**	137.38 ± 12.50	136.72 ± 8.75	0.734
**Birth weight (g)**	3197.85 ± 423.95	3187.75 ± 462.37	0.896
**Birth length (cm)**	49.94 ± 2.06	49.57 ± 2.15	0.329
**Head circumference (cm)**	34.27 ± 1.40	34.28 ± 1.35	0.981
**^a^ Apgar 1**	8.81 ± 0.63	8.71 ± 1.01	0.527
**^a^ Apgar 5**	9.93 ± 0.26	9.80 ± 0.58	0.104
**Umbilical cord Ph**	7.24 ± 0.07	7.23 ± 0.08	0.610
**Weight at hospital discharge (g)**	3003.33 ± 417.94	2976.87 ± 466.89	0.747
**Sex (n/%)**			
Male	29/42.0	37/52.9	
Female	40/58.0	33/47.1	0.201
**Gestational age of delivery (n/%)**			
Full term ≥ 37 wk	63/94.0	61/88.4	
Preterm < 37 wk	4/6.0	8/11.6	0.248
**Stratified birth weight (n/%)**			
Low < 2500 g	3/4.3	10/14.3	
Adequate (2500–4000 g)	64/92.8	54/77.1	
Macrosomia (>4000 g)	2/2.9	6/8.6	0.037
**NICU (n/%)**			
No	60/87.0	50/71.4	
Yes	9/13.0	20/28.6	0.024

^a^ APGAR appraises the newborn’s heart rate, muscle tone, reflexes, color, and respiratory effort [63].

**Table 4 jcm-11-04045-t004:** Birth weight (BW) differences related to maternal type of gestational weight gain (GWG) between the IG and CG.

		IG (n = 69)	CG (n = 70)	*p*-Value
**BW based on the type of GWG** **(g)**	**Inadequate**	3085.86 ± 458.620	3052.60 ± 477.092	0.807
	**Adequate**	3279.29 ± 381.807	3285.19 ± 500.401	0.964
	**Excessive**	3504.29 ± 371.028	3207.86 ± 389.238	0.088
	***p*-value**	0.035	0.261	

## Data Availability

The data are not publicly available due to the agreement between the university and participant hospitals.

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
