# Peer review of "A Virtual Exercise Program throughout Pregnancy during the COVID-19 Pandemic Modifies Maternal Weight Gain, Smoking Habits and Birth Weight—Randomized Clinical Trial"

_jcm, 2022, doi:10.3390/jcm11144045_

Round 1
Reviewer 1 Report
The authors have taken a very important aim to assess the effectiveness of a virtual exercise program implemented during pregnancy. This is especially important in the post-pandemic period, showing the possibilities of using online exercise programs, not only in conditions of compulsory social isolation, but also to generally increase the availability of exercise for pregnant women.
The study was conducted in accordance with high standards of scientific research methodology, the results should certainly be published, however, before publication, this manuscript requires some important changes.
Based on the presented data, I recommend to change the title for “A virtual exercise program throughout pregnancy during the COVID-19 pandemic modifies maternal weight gain and birth outcomes. Randomized clinical trial”.
Certainly, the issue of changing smoking habits through the exercise intervention has not been properly analyzed and presented in this manuscript.
First, in the abstract the authors stated that “smoking during pregnancy was found more often in the CG (12/17.1%) compared to the IG (3/4.4%) (p = 0.016).” But (as it was presented in the results), it was at baseline, so the significant difference is not an effect of intervention, but appeared by chance after randomization of the participants to the two groups. So, taking into account that the authors planned to use “smoking” as a main variable to be analyzed, this effect of randomization makes a serious methodological bias. There should be the same rate of smokers and non-smokers to start the analysis. Or at least, the analysis of post-intervention changes should be adjusted to the baseline difference between groups.
Secondly, the authors mentioned “smoking” as secondary outcome (point 2.4.2). Please change it for the primary outcome if you would like to keep the title.
Thirdly, the authors presented neither the methodology how to prove that “the smoking during pregnancy would be modified due to the virtual exercise program” nor the data to support this thesis (only the baseline participants responses on whether they smoked before and during pregnancy and the correlations between smoking and selected birth outcomes). The authors stated that (lines 324-326) “the current situation caused by COVID-19 is perceived by pregnant individuals as stressful [75,76] and even though in our study all received information on healthy lifestyle (information on smoking included), the CG maintained their smoking habits.” However, these data were not presented. Did the authors use any questionnaire for this variable (changes of smoking habits)? If yes, please present both in the methodology and results sections.
In my opinion, the authors didn’t justified in this manuscript what they stated in the discussion (lines 333-336) (at least it is not present in this manuscript) in the discussion “our RCT confirms the beneficial effects of exercise during pregnancy and demonstrates the importance of using lifestyle-focused treatments as a necessary factor for the prevention of unhealthy habits such as smoking” (the second part of this sentence related to the maternal weight gain was justified).
Therefore, this manuscript requires some modifications to make it more consistent. After removing “smoking” variable from the title, I recommend to cut some parts about smoking from the introduction and focusing mainly on maternal weight gain and birth outcomes. I do consider this data is very valuable and important, so after some revision of this manuscript, certainly it should be published.
Minor comments:
1. Please improve the references:
Line 181: the 2009 Institute of Medicine (IOM) guidelines 52 (REF).
Line 294: It appears that 292 being active helps to control low birth weight [56] and reduces the risk of macrosomia by 293 up to 30% 33 (REF).
2. Please use the superscript for m2 : Line 175 (and further): BMI was calculated as weight (kg) divided by height (m)2.
Author Response
Thank you very much for your time and effort in reviewing our article.
First, an English speaking co-author checked the document for grammar and spelling mistakes.
Secondly, here are the items reviewed:
The authors have taken a very important aim to assess the effectiveness of a virtual exercise program implemented during pregnancy. This is especially important in the post-pandemic period, showing the possibilities of using online exercise programs, not only in conditions of compulsory social isolation, but also to generally increase the availability of exercise for pregnant women.
The study was conducted in accordance with high standards of scientific research methodology, the results should certainly be published, however, before publication, this manuscript requires some important changes.
Based on the presented data, I recommend to change the title for “A virtual exercise program throughout pregnancy during the COVID-19 pandemic modifies maternal weight gain and birth outcomes. Randomized clinical trial”.
We have modified the title but feel that smoking habits should remain in the title based on the explanation below.
Certainly, the issue of changing smoking habits through the exercise intervention has not been properly analyzed and presented in this manuscript.
First, in the abstract the authors stated that “smoking during pregnancy was found more often in the CG (12/17.1%) compared to the IG (3/4.4%) (p = 0.016).” But (as it was presented in the results), it was at baseline, so the significant difference is not an effect of intervention, but appeared by chance after randomization of the participants to the two groups. So, taking into account that the authors planned to use “smoking” as a main variable to be analyzed, this effect of randomization makes a serious methodological bias. There should be the same rate of smokers and non-smokers to start the analysis. Or at least, the analysis of post-intervention changes should be adjusted to the baseline difference between groups.
Smoking was recorded prior to pregnancy and throughout pregnancy (recorded at all hospital visits up to the one prior to labour). For the calculation, data from smoking at the last visit (just before labour) was used, since no woman had started smoking during pregnancy. We added the following to clarify in the methods (Line 175-178).
Smoking status was determined from medical records. The participants were asked if they smoked prior to pregnancy (Yes/No) and at all obstetric visits (Yes/No). Smoking status during pregnancy was represented by the yes/no response at the last visit (just before labour).
From these data, there are no significant differences in smoking before pregnancy, but there are differences between groups during pregnancy represented by the last data point before delivery, so the program would have an "effect" on this variable.
Similarly, following your comment and to provide clarity to the text, “at baseline” was removed from the title of table 1 to avoid confusion, since the record of smoking during pregnancy is also shown in this table.
Secondly, the authors mentioned “smoking” as secondary outcome (point 2.4.2). Please change it for the primary outcome if you would like to keep the title.
The study was designed to examine birth weight as the primary outcome. We were interested also in smoking habits and maternal weight gain which were considered as secondary outcome analyses. We feel that it would not be appropriate to change the primary outcome at this point but feel strongly that based on the interesting outcomes for the secondary analysis (smoking and maternal weight gain), these should remain in the title. We highlight the important and novel aspects of our analysis in the title. This would be beneficial to understand the whole context of the study.
Thirdly, the authors presented neither the methodology how to prove that “the smoking during pregnancy would be modified due to the virtual exercise program” nor the data to support this thesis (only the baseline participants responses on whether they smoked before and during pregnancy and the correlations between smoking and selected birth outcomes). The authors stated that (lines 324-326) “the current situation caused by COVID-19 is perceived by pregnant individuals as stressful [75,76] and even though in our study all received information on healthy lifestyle (information on smoking included), the CG maintained their smoking habits.” However, these data were not presented. Did the authors use any questionnaire for this variable (changes of smoking habits)? If yes, please present both in the methodology and results sections.
Clarifications have been provided in the text on lines (141-142 and 148-149 methods) that both groups received the same type of information on healthy habits, including advice and information on smoking during pregnancy. With this, having the same access to lifestyle information, it is still observed that in the IG women smoked less during pregnancy, so the program appears to be effective.
In my opinion, the authors didn’t justified in this manuscript what they stated in the discussion (lines 333-336) (at least it is not present in this manuscript) in the discussion “our RCT confirms the beneficial effects of exercise during pregnancy and demonstrates the importance of using lifestyle-focused treatments as a necessary factor for the prevention of unhealthy habits such as smoking” (the second part of this sentence related to the maternal weight gain was justified).
As previously explained, both groups received information for a healthy pregnancy. All individuals received information regarding smoking during pregnancy and it can be speculated that giving information regarding smoking is beneficial, but we also saw a decline in maternal smoking in the intervention group compared to the control group suggesting education accompanied by an exercise program may be very beneficial to assist in smoking cessation. More research is needed to confirm this interesting observation.
Therefore, this manuscript requires some modifications to make it more consistent. After removing “smoking” variable from the title, I recommend to cut some parts about smoking from the introduction and focusing mainly on maternal weight gain and birth outcomes. I do consider this data is very valuable and important, so after some revision of this manuscript, certainly it should be published.
Based on the above comments, we feel that discussion of smoking should remain in the introduction and also the discussion. Please see comments from Reviewer #2.
Minor comments:
- Please improve the references:
Line 181: the 2009 Institute of Medicine (IOM) guidelines 52 (REF)
Following the reviewer's comment, the reference has been improved
Line 294: It appears that 292 being active helps to control low birth weight [56] and reduces the risk of macrosomia by 293 up to 30% 33 (REF).
Following the reviewer's comment, the reference has been improved
Please use the superscript for m2 : Line 175 (and further): BMI was calculated as weight (kg) divided by height (m)2.
Following the reviewer's comment, the wording has been improved
Reviewer 2 Report
Manuscript type: Article
Journal JCM (ISSN 2077-0383)
Manuscript jcm-1784677
Title
A virtual exercise program throughout pregnancy during the COVID-19 pandemic modifies maternal smoking habits and birth weight. Randomized clinical trial.
Comments, observations and Suggestions for authors.
The manuscript, by dr. SILVA-JOSE and colleagues, aimed to evaluate the influence of a supervised virtual exercise program throughout pregnancy on birth weight, while the influence of virtual exercise and the relationship between changes in the maternal environment during pregnancy and other neonatal outcomes were evaluated, as well.
Please see below are some suggestions, which would allow the article to be improved for publication in Journal of Clinical Medicine.
1. The experimental design is well performed. Figures/table are informative. Scientific writing is adequate.
2. In order to improve the reading, unnecessary spaces between paragraphs sohuld be removed. E.g., lines 40-41, 49-50, 56-57, 66-67, 80-81, 85-86, 90-91, 272-279, 275-276, 288-289 etc…
3. Acronyms should be carefully checked for a better reading and reported with their complete name when mentioned for the first time. For instance, line 76, COVID-19 should be Coronavirus disease 2019 (COVID-19), line 96 NICU should be Neonatal intensive care unit (NICU) etc…
4. Lines 44-43 additional long-term metabolic effects of high birth weight can be found here DOI: doi: 10.3389/fped.2021.675775
5. Lines 57-66 parental diet can extensively alter epigenetic pattern, in terms of DNA methylation, of certain gene during gametogenesis. This phenomenon can alter the neonatal weight as well as influence imprinting marks in the offspring (PMID:34368137). These modifications have been hypothesized to be transferred through generations, potentially affecting the health of adult offspring (PMID:34368137). This important information and supporting references should be, at least briefly, included as a background
6. Lines 57-66 I suggest also including a couple of words on the negative effect of alcohol consumption upon birth weight DOI: 10.3109/00016349109007877 and doi:10.1001/jama.1984.03350140021018
7. Lines 62-64 The negative effects of cannabis smoking sohuld also be mentioned doi: 10.5694/mja2.50624
8. Line 91, better current COVID-19 pandemic
9. The 2.4 paragraph can be condensed in a unique paragraph, thus without subsections
10. 2.5 section please include supporting references
11. Line 231 please remove the period before “(Figure 1).
12. Table 3, The meaning of Apgar should be detailed, it would be helpful for a non-expert reader
13. Line 281 “main objective of the present study was…”
14. Line 289 better “our results”
15. Line 294 please revise this typo error “30% 33 (REF).”
16. Line 322 among negative pregnancy outcomes, I suggest mentioning spontaneous abortion DOI: 10.1080/00016340600589560
Author Response
Thank you for your time and effort in reviewing the article.
First, an English speaking co-author checked the document for grammar and spelling mistakes.
Secondly, here are the comments on the revised points:
The manuscript, by dr. SILVA-JOSE and colleagues, aimed to evaluate the influence of a supervised virtual exercise program throughout pregnancy on birth weight, while the influence of virtual exercise and the relationship between changes in the maternal environment during pregnancy and other neonatal outcomes were evaluated, as well.
Please see below are some suggestions, which would allow the article to be improved for publication in Journal of Clinical Medicine.
The experimental design is well performed. Figures/table are informative. Scientific writing is adequate.- In order to improve the reading, unnecessary spaces between paragraphs should be removed. E.g., lines 40-41, 49-50, 56-57, 66-67, 80-81, 85-86, 90-91, 272-279, 275-276, 288-289 etc…
These spaces have been corrected
- Acronyms should be carefully checked for a better reading and reported with their complete name when mentioned for the first time. For instance, line 76, COVID-19 should be Coronavirus disease 2019 (COVID-19), line 96 NICU should be Neonatal intensive care unit (NICU) etc…
Following the reviewer's comment, the full names of the requested abbreviations have been added and the entire document has been revised.
Lines 44-43 additional long-term metabolic effects of high birth weight can be found here DOI: doi: 10.3389/fped.2021.675775
Following the reviewer's comment, this information has been added to the document. (Lines 44-46)
- Lines 57-66 parental diet can extensively alter epigenetic pattern, in terms of DNA methylation, of certain gene during gametogenesis. This phenomenon can alter the neonatal weight as well as influence imprinting marks in the offspring (PMID:34368137). These modifications have been hypothesized to be transferred through generations, potentially affecting the health of adult offspring (PMID:34368137). This important information and supporting references should be, at least briefly, included as a background.
Following the reviewer's comment, this information has been added to the document. (Lines 67-73)
- Lines 57-66 I suggest also including a couple of words on the negative effect of alcohol consumption upon birth weight DOI: 10.3109/00016349109007877 and doi:10.1001/jama.1984.03350140021018
Following the reviewer's comment, this information has been added to the document. (Lines 67-73)
- Lines 62-64 The negative effects of cannabis smoking should also be mentioned doi: 10.5694/mja2.50624
Following the reviewer's comment, this information has been added to the document. (Lines 67-73)
- Line 91, better current COVID-19 pandemic
Following the reviewer's comment, the text has been modified
- The 2.4 paragraph can be condensed in a unique paragraph, thus without subsections
Following the reviewer's comment, the paragraph has been condensed
2.5 section please include supporting references
Following the reviewer's comment, references to the randomization process have been added.
Line 231 please remove the period before “(Figure 1).
Following the reviewer's comment, the period has been removed
Table 3, The meaning of Apgar should be detailed, it would be helpful for a non-expert reader
Based on reviewer comment, this information has been added below the table and the scientific reference has been provided. (Line 257)
Line 281 “main objective of the present study was…”
Following the reviewer's comment, the text has been added. (Line 285)
14. Line 289 better “our results”
Following the reviewer's comment, the text has been modified (293)
Line 294 please revise this typo error “30% 33 (REF).”
Based on reviewer comment, this typo error has been checked and corrected
- Line 322 among negative pregnancy outcomes, I suggest mentioning spontaneous abortion DOI: 10.1080/00016340600589560
Following the reviewer's comment, this important reference has been added (line 326)